# Community-Scale Rural Drinking Water Supply Systems Based on Harvested Rainwater: A Case Study of Australia and Vietnam

**Tara T. Ross, Mohammad A. Alim and Ataur Rahman \***

School of Engineering, Design and Built Environment, Western Sydney University, Locked Bag 1797, Penrith 2751, Australia; 17149192@student.westernsydney.edu.au (T.T.R.); m.alim@westernsydney.edu.au (M.A.A.)
\* Correspondence: a.rahman@westernsydney.edu.au; Tel: +61-02-47360-145

**Abstract:** Rainwater harvesting (RWH) systems can be used to produce drinking water in rural communities, particularly in developing countries that lack a clean drinking water supply. Most previous research has focused on the application of RWH systems for individual urban households. This paper develops a yield-after-spillage water balance model (WBM) which can calculate the reliability, annual drinking water production (ADWP) and benefit–cost ratio (BCR) of a community-scale RWH system for rural drinking water supply. We consider multiple scenarios regarding community aspects, including 150–1000 users, 70–4800 kL rainwater storage, 20–50 L/capita/day (LCD) drinking water usage levels, local rainfall regimes and economic parameters of Australia (developed country) and Vietnam (developing country). The WBM analysis shows a strong correlation between water demand and water supply with 90% system reliability, which allows both Australian and Vietnamese systems to achieve the similar capability of ADWP and economic values of the produced drinking water. However, the cost of the Vietnamese system is higher due to the requirement of larger rainwater storage due to larger household size and lower rainfall in the dry season, which reduces the BCR compared to the Australian systems. It is found that the RWH systems can be feasibly implemented at the water price of 0.01 AUD/L for all the Vietnamese scenarios and for some Australian scenarios with drinking water demand over 6 kL/day.

**Keywords:** rainwater harvesting; water balance model; rural community; drinking water production; benefit–cost ratio; water price

## 1. Introduction

According to UNESCO, approximately half of the global population lives in rural areas and the majority are categorised as the low-income group. This condition is dire in Asia and Africa. It is estimated that almost 900 million people lack access to essential water supply. As a result, these regions face an extraordinary human health crisis, let alone the effects on economy [1]. The advanced urban water supply system is not feasible for rural areas due to overall smaller water demand. To meet the United Nations' Sustainable Development Goals (UNSDG), which is universal and equitable access to safe drinking water at an affordable cost, identifying alternative sustainable water resources is crucial [2]. Rainwater harvesting (RWH) may be a potential solution to mitigate the problem [3]. Many studies have shown the advantages of RWH in reducing water demand from mains in urban areas [4–6]. These studies proposed the use of harvested rainwater for non-potable purposes. However, research on drinking water production based on harvested rainwater for rural communities is limited in the literature.

RWH has been an active research area to mitigate impending water issues stemming from rising mains water demand and increasing water scarcity. Surface water and groundwater are overexploited and often polluted at many locations globally. The United

Nations Environment Program (UNEP) highlighted the increased uptake and benefits of RWH, especially in rural areas lacking access to clean drinking water [7]. The UNSDG report found that approximately 30% of the global population living in rural areas have limited access to safe drinking water. Thus, there is an urgent need to design affordable and sustainable RWH systems to meet the clean water goals in the UNSDG programs [8].

RWH has been practiced over thousands of years [9], and is increasingly being adopted in recent years [10]. Many developed countries have well-established RWH policies and regulations in place [11], and have made RWH systems mandatory for new development areas [12–14]. RWH systems are becoming popular in developing countries such as Bangladesh, Brazil and Malaysia, but mainly in cities and urban areas [15–17]. In fact, urban areas have widely installed regulated RWH systems to reduce pressure on mains for domestic, commercial, industrial and agricultural purposes [10]. However, most of the poor rural areas are unable to install RWH systems at the individual level due to the higher installation cost of RWH systems and affordability [18,19]. Moreover, climate change associated with increasing temperatures, and shifting rainfall patterns and frequent floods and droughts are raising concerns about the sustainability of the rural domestic water supplies [20].

Several studies have focused on drinking water production based on harvested rainwater for single dwellings in urban areas [21,22]. However, RWH studies for drinking water production in rural areas at community levels are quite limited or insufficiently documented or generally carried out empirically [23]. For example, a Malaysian large-scale RWH system with non-potable water usage of 160 kL/day serving 600 people revealed a significant water savings of 58% [24]. An Australian small-scale system considered a 5–15 kL storage tank with drinking water usage of 15 L/day/capita and a maximum daily filtration capacity of 500 L/day serving around 33 people [25]. Vietnamese RWH systems at three sites in northern Vietnam considered storage tanks 8–10 kL serving 100–500 people with drinking water usage of 1–3 L/day/capita [26]. Similarly, the potential of village-scale RWH systems to help rural areas where individuals cannot afford their own system was recognised by Amos et al. in 2020 [27]. Those studies have proven that the communal approach of RWH can provide a reliable supply of domestic water.

Village-scale RWH systems with community reservoirs have been adopted in Sri Lanka, India and around the world to provide safe water [28]. Moreover, RWH has been acknowledged as an exclusive approach to domestic water supply on many islands and in several archipelagos of the Mediterranean Sea due to other non-conventional water supply methods based on the use of water tankers and/or salination having high operational costs in the medium to long term [29]. Harvested rainwater is an excellent source of drinking water because of the low contaminant level, which can be removed easily by simple treatment. This makes RWH a welcoming solution in water-scarce regions such as geogenic unfavourable areas, affected by higher arsenic and fluoride [30]. In addition, the concern about low mineralisation in harvested rainwater can be balanced by proper daily diet and/or additional minerals in the rainwater treatment process, as noted by Amos et al. (2020), and has been practiced over many years in certain parts of the globe [27].

Moreover, regarding the quality of harvested rainwater, a literature review has shown that many rural areas in both developed and developing countries with limited/no access to mains water supply adopt traditional and unsafe RWH practices; for example, most of the villagers use harvested rainwater for potable purposes without formal treatment [31,32]. In fact, rural residents are evidently exposed to heavy metals in harvested rainwater [33]. Moreover, the presence of various opportunistic and pathogenic microorganisms has been confirmed in harvested rainwater sources, which means the microbial quality of harvested rainwater does not always comply with drinking water guidelines [34]. Therefore, rainwater treatment strategies including filtration, metal/chemical additives, chlorination, ozone, UV, solar disinfection and solar pasteurisation have been used for harvested rainwater around the world. The

combination of treatment methods appeared to be essential and more effective in ensuring the quality of harvested rainwater [35]. However, little effort has been made to develop a cost-effective combination of rainwater treatment strategies for developing countries, where these technologies are needed the most. It is important for future research to investigate the rainwater combination treatment methods in the rural community-scale RWH systems. Hence, this study investigates the practical aspects and the optimal conditions required for the successful application of the community-scale drinking water supply system based on harvested rainwater within a range of rainfall regimes and economic scenarios of rural areas.

Australia and Vietnam are good examples of contrasting study locations with different RWH practices. Australia is the driest inhabited continent, where freshwater is a precious commodity. Australia suffers from severe droughts, catastrophic bush fires and water restrictions, which has that boosted the adoption rate of RWH systems to 34% [36]. The use of RWH systems has become an integral part of urban development within Australia, which has benefitted urban water supplies by supplementing mains water and reducing stormwater discharge into sewerage systems and pollution of bodies of freshwater [37]. However, there is an ongoing challenge of adopting RWH to provide secure drinking water supplies in rural areas that have limited/no access to mains and where drinking water is costly [23]. In contrast, despite Vietnam receiving abundant rainfall, rural Vietnamese households have shortfalls in rainwater collection for potable use throughout the year owing to unaffordable storage measures and inadequate technology. In fact, the percentage of rural Vietnamese households that used RWH for potable purposes was 20% and for drinking only was 67%. As a result, the remaining consumers were supplied with alternative unimproved or unmonitored groundwater and bottled water [32,38]. Thus, there has been a renewed interest from Vietnam authorities in using RWH for rural drinking water supplies [26,39,40]. The specific objective of this study is to examine the feasibility of rural community-scale RWH systems under various plausible scenarios in two different rainfall regimes (Australia—developed country, and Vietnam—developing country). It is expected that the outcomes of this study will assist in adopting an RWH system as a solution to the drinking water problem in rural areas.

## 2. Literature Review

This literature review focuses on fundamental conceptional design of a community-scale RWH system. The basic components of a drinking water supply unit based on roof catchment fed RWH system consists of roof catchment itself plus collection, storage, treatment and distribution components. Conceptual design of the RWH system including all the components is an important aspect in the planning of this project for future water supply. Although advancements to support these RWH system components are available, most of the designs were focused on individual household based RWH systems and there are limited technology applications for community-scale RWH systems. By constructing large-scale rooftop rainwater collection systems and community tank, villagers can have a reliable water supply. Different types of traditional rainwater collection systems (e.g., percolation system, narrow open well, earthen check dam, village pond, stairs well system, etc.) are still being used in rural areas of many developing countries such as Sri Lanka, India and Bangladesh [28]. A recent research work has proposed an innovative solution called directional tunnelling using visual Building Information Modelling (BIM) tools to support the activity of RWH for small rural communities [41]. Such visual modelling tools can be useful in the conceptual design phase of the proposed RWH systems.

The main goal of designing the proposed RWH system is to balance the social, economic and environmental aspects so that water demand is met, the system cost-benefit is balanced in the investment timeframe, and the environment is protected and benefited from. The technical structure of each component plays an important role to build the proposed drinking water supply system. The storage tank size would depend on several

factors such as water demand, rainfall availability, losses and system reliability [42]. Therefore, in case of community-scale rainwater storage, mega structure of the storage tank or reservoir would be required. However, developing awareness of the sustainability concept, several innovative storage ideas in the RWH industry such as bladder tanks and food-grade polymer liners can be applied for the community rainwater storage tanks or reservoirs. This needs further research to determine the cost-benefit of such large storage systems.

Regarding the components of rainwater treatment, it is crucial that public health issue is considered when developing efficient rainwater treatment strategies [43]. Compared to surface water and groundwater resources, rooftop harvested rainwater often lacks essential minerals (e.g., sodium, potassium, fluoride, etc.) and is contaminated by heavy metals, organic matters and pathogenic microorganism [44,45]. The most common techniques of the rainwater treatment include filtrations from simple low-tech such as slow sand filtration to high-tech membrane technologies (e.g., microfiltration, ultrafiltration, nanofiltration, reverse osmosis, etc.), disinfection (e.g., chlorination, ultraviolet radiation (UV), solar disinfection and solar pasteurisation, etc.) and chemicals/minerals addition [46,47]. Varying degrees of treatment efficiency of these technologies would be de-pendent on cost, ease of management and social acceptability. The combination of simple filtration with disinfection and chemical additives has been proven to be more effective for harvested rainwater to meet the physical, chemical and microbiological aspects of drinking water guidelines [35]. Moreover, the type of granular settlement filter that has multiple layers of materials (e.g., sand, gravel, charcoal/activated carbon, crushed recycled glass, etc.) are compacted in a filtration unit/chamber, which has been found to be simple, low-cost and efficient to remove contaminants from harvested rainwater for drinking purposes [48–51]. Amongst disinfection strategies, UV technology appeared to be simple, cost-effective with minimal operational maintenances. The strategy of combining granular activated carbon with UV appeared to be a suitable solution for rainwater treatment in rural communities in developing countries where resources are limited [52,53], which can be considered for the conceptual design of the treatment component of the proposed system.

It has been noted that central treatment procedure that was used in the secondary treatment process of a water treatment plant appeared to be practical in a community-scale water supply system based on rainwater-stormwater management [54]. Similarly, the type of central treatment process is suggested to apply for the proposed system. However, the design specification of the central treatment process would incorporate necessary structural subassemblies involving the common rainwater treatment techniques as mentioned above".

## 3. Materials and Methods

### 3.1. Study Areas and Data Selection

Australia and Vietnam were selected as our study areas. The purpose of using rainfall data from different countries at multiple locations in this study is to compare local and global implementation aspects of the RWH systems at community scale.

Australia receives high rainfall along its coastal regions, where over 84% of the population resides. The northern half of the continent typically experiences a monsoonal summer wet rainfall season from October to April, with the rest of the year remaining relatively dry. The southern half of the continent, particularly South Australia (SA), Victoria (VIC), Tasmania (TAS), part of New South Wales (NSW) and southern Western Australia (WA), experiences its highest rainfall during the cooler months from November to April [55]. In addition, Australia has witnessed a major change in its rainfall patterns in the last 50 years. Rainfall has increased in the Northern Territory (NT) and northern WA, whilst decreased in southeast Australia [56]. NSW was selected for this study, which has a mean annual rainfall of about 800–1200 mm.

Vietnam is a developing country located in a hazard-prone area, affected by frequent floods and storms [57]. The country is in the monsoonal Southeast Asia region with a mean annual rainfall between 1400–2400 mm. The rainy season in Vietnam accounts for about 80–90% of the annual rainfall [58]. The rainy season of northwest, northeast and north Delta Vietnam starts in April–May with a peak in July–August and ends in September–October, while the southern Vietnam rainy season begins in May, peaks in September and ends in November [59]. Southern Vietnam's rainfall pattern was more uniform than other parts of Vietnam, which was used as the study area.

Australian historical daily rainfall data were obtained from the Australian Bureau of Meteorology (BOM), including three NSW rainfall stations: Bilpin, Springwood and Richmond (Figure 1, Table 1). Vietnam's historical daily rainfall data were collected from the Southern Regional Hydro-meteorological Centre (SRHC), including three southern Vietnam rainfall stations: Thu Dau Mot (TDM), Nha Be (NB) and Tan Son Hoa (TSH) (Figure 1, Table 1). The selected Australian rainfall stations had a data length of 60 years (1960–2019), while the selected Vietnam rainfall stations contained 40 years of data (1980–2019) with no rainfall records during the Vietnam War (1954–1975). The average monthly rainfall of the selected Australian stations was between 100–150 mm in the rainy season (Oct–Apr) and varied by around 49 mm during the dry season (Jul–Sep). The monthly rainfall of the selected Vietnamese stations was extremely low (below 19 mm) in the dry season (Dec–Mar) but was extremely high (around 200–300 mm) in the rainy season (May–Oct) (Figure 2). Since the water savings and reliability of a rainwater tank depend on the total annual rainfall, the average rainfall in dry months and monthly water demand, the results of our analysis are affected by these rainfall characteristics of Australia and Vietnam.

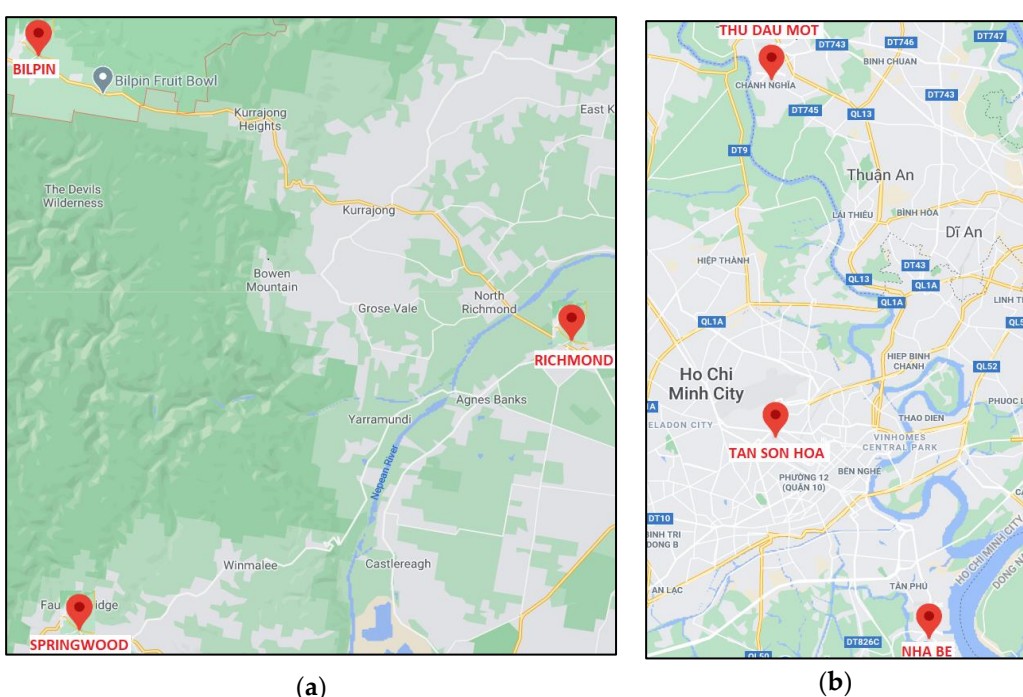

(**a**)      (**b**)

**Figure 1.** Locations of selected rainfall stations. (**a**) Australia rainfall stations: Bilpin, Richmond and Springwood; (**b**) Vietnam rainfall stations: TDM, TSH and NB. Source: Google Maps www.google.com/maps (accessed: 2 January 2022).

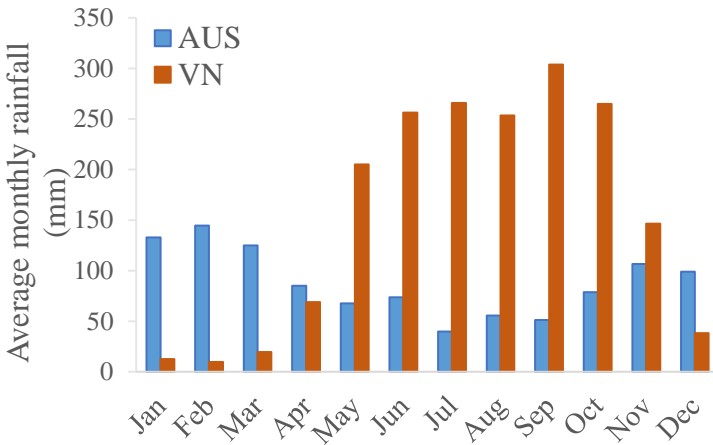

**Figure 2.** Average monthly rainfall of the selected stations in Australia and Vietnam.

**Table 1.** Summary of rainfall data of the selected stations in Australia and Vietnam.

| Countries | Rainfall Stations | Rainfall Record Periods | Annual Rainfall (mm) | Annual Rainy Days | Annual Rainfall (mm) |
|---|---|---|---|---|---|
| | Bilpin | 1960–2019 | 1303 | 133 | |
| Australia | Richmond | 1960–2019 | 821 | 119 | 1059 |
| | Springwood | 1960–2019 | 1055 | 108 | |
| | TDM | 1977–2019 | 1891 | 151 | |
| Vietnam | NB | 1980–2019 | 1685 | 140 | 1844 |
| | TSH | 1980–2019 | 1955 | 159 | |

*3.2. Design of Community-Scale Drinking Water Supply System Based on Harvested Rainwater*

The design methodology to develop the proposed system is based on the problem-solving strategies established by Samuel and Wier in 1999 [60] that is summarised hereunder.

Problem identification: The aim and purpose of the proposed system is to provide a reliable, low-cost and eco-friendly drinking water supply at community scale by using rooftop harvested rainwater. Hence, social, economic and environmental aspects have been taken into consideration to develop the system including: (i) the domestic water demand of the rural community is satisfied by the water supply system, (ii) the water produced by the system must meet local and international drinking water quality guidelines, (iii) the financial benefit can compete or defeat all costs involved in the life cycle of the system, (iv) the system can help the environment by reducing carbon footprint, floods and pollutant wash-off.

Chosen approach: To address the above problem, this study proposes a modelling tool to analyse the system reliability and life cycle cost–benefit (Section 3.3). Moreover, to ensure that the system can produce safe drinking water, the technical components of the rainwater treatment system are selected, which considered physical, chemical and microbiological aspects as per the Australian and WHO drinking water guidelines. Finally, we considered the use of a solar energy system as the eco-friendly components of the system.

Chosen products and technologies: The proposed RWH system consists of rainwater collection apparatuses on rooftops in the selected community, a large rainwater storage tank, a rainwater treatment system, a small tank for daily drinking water production, a solar energy system, water pressure pumps, drinking water distribution pipelines and individual household dispensers or taps (Figure 3). The rainwater treatment options

selected in our study are commonly adopted methods, which includes pre-treatment for removing physio-chemical contaminants and disinfection for eliminating harmful pathogens [21,35,47,48,61]. The design of the community-scale RWH system includes a large rainwater storage tank and a small storage tank for fresh daily treated rainwater, so-called produced drinking water, which is delivered to the users within 24 h. The installation of the RWH system is generally offered free of charge by suppliers. Plumbing works related to RWH collection and drinking water distribution require licensed plumbers. In the selected area, training of local plumbers will be necessary so that they understand this new RWH system. Moreover, the installation of boundary fences to protect the RWH system is included in the labour costs. The designed components of the proposed RWH system that contribute to the estimation of capital cost, replacement cost and maintenance cost in the life cycle analysis are described below.

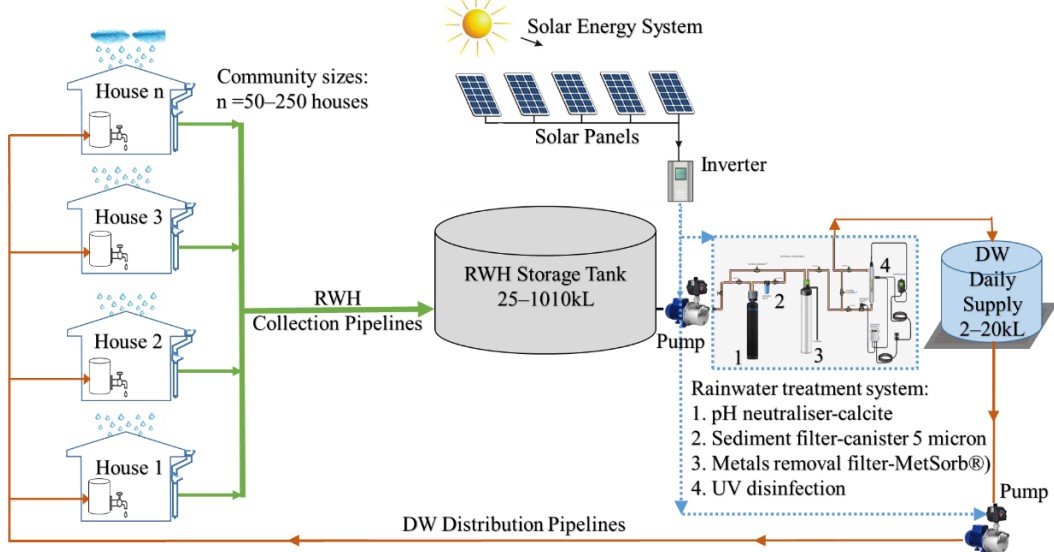

**Figure 3.** Illustration of community-scale RWH system for drinking water supply.

a.   *Rainwater collection apparatus*

In good condition, existing household roofing materials (e.g., zincalume, aluminium and steel) can be used as catchment areas for rainwater collection. Leaf eater, first flush diverter, downpipe and plumbing accessories should be used for each roof catchment to prevent debris and contaminants from entering the storage tank. Food-grade polyvinyl chloride (PVC) or high-density polyethylene (HDPE) collection pipelines should be used to transport rainwater from rooftops around the selected community houses to the storage tank. The estimated average pipeline distance required for each house is 60 m, including a 20 m distance of collection branch from a property to the main collection pipe and a 40 m distance from one to the next property [62]. The products used in this section will be supplied by local plumbing industries [63].

b.   *Rainwater storage tank*

The rainwater storage tank can be sourced by local tank industries, such as the Australian product Rhino rural rainwater tank series, which consists of premanufactured corrugated steel (zincalume) panels with standard inclusions and optional extras (e.g., fire fittings required by local governments, tank level gauge, dust and vermin proofing seal). Multiple tanks with a capacity of 25–365 kL can be connected depending on the required size. The tanks can be installed at the site by the supplier within a short period, e.g., 24 h. The tank installation will require a pad layer of crushed dust or sand, an exclusion zone of blue metal stone, and bulk water of 10% tank capacity to stabilise the tank liner upon the completion of the tank installation [64].

c. *Multi-step rainwater treatment system*

Step 1: It is reported that roof rainwater is likely to be acidic, with a pH < 6.0 due to the leaching of roofing materials and atmospheric pollution [65]. A pH neutraliser will be employed in step 1 of the rainwater treatment system to neutralise the rainwater's acidity to meet the drinking water guideline of pH 6.5–8.5 [66]. The blend of calcite (Calcium Carbonate, $CaCO_3$) with corosex (Magnesium oxide, MgO) 80:20 or 90:10 may be considered for this purpose if rainwater pH < 6.0. Otherwise, calcite media can be used alone if pH > 6. Acidic rainwater will slowly dissolve $CaCO_3$ and MgO to raise pH levels upon contact with the media. One of the advantages of calcite media is its self-limiting properties, stabilising the pH to reach a non-corrosive equilibrium. Additionally, an automated valve for regular backwash and rinses at a preferred time interval (weekly) will help reclassify the calcite media bed, maintain high service flow rates (9.6 $m^3$/h or 160 L/min) and remove suspended solids, which otherwise can become compacted. Moreover, depending on pH levels, water chemistry and service flow, the calcite bed (calcite media 50–150 kg) will require periodical (annual) replenishment as calcite is depleted. This type of pH neutraliser is commonly available at most local water treatment suppliers around the world [67].

Step 2: Rainwater turbidity > 1 NTU can hinder the rainwater treatment process. To control rainwater turbidity, several mesh barriers such as leaf eaters, first flush diverters and strainer baskets should be placed prior to the inlet of the storage tank. In addition, a canister pleated a washable sediment filter will be employed in step 2. The pleats in this filter are designed to be washed and reinstalled, which is environmentally friendly and more economical than other similar single-use filters. Moreover, quarterly washes and annual replacement are required to remove odours and bacterial contaminants to minimise unexpected problems in subsequent treatment steps. Additionally, the 5-micron filter is suited to sediment, dirt and algae removal, which is an effective filtration for pre-filter purposes at a high service flow rate (132 L/min) with no reduction in water pressure. Importantly, this type of poly filter is common and available at local suppliers of water filters in both developed and developing countries [68].

Step 3: Roof rainwater is likely to be contaminated with metals due to the leaching of metal roofing materials. Therefore, a heavy metals removal filter will be employed at this stage of treatment. For example, MetSorb® is an excellent filter media that has increased the surface area afforded by Titanium coupled with advanced pore volume. The filter provides fast adsorption kinetics for heavy metals at a high service flow rate (160 L/min). Moreover, the filter media maintains a higher adsorbent capacity and a lower ion interference than other competitive products such as activated carbon, silica gel, iron and alumina-based adsorbents. MetSorb® will be able to remove odorous compounds and a wide variety of heavy metals (e.g., Arsenic, Lead, Cadmium, Copper, Chromium, Selenium and Zinc) from aqueous sources to meet drinking water guidelines. Additionally, the filter media is non-hazardous and disposable as solid waste, and comes with a 10-year warranty. A simple automated periodic (monthly) backwash technology will be included to keep the media clean and operating efficiently without any maintenance [69].

Step 4: Microbial contamination of roof harvested rainwater via atmospheric deposition, leaching and weathering of roof materials, faecal contamination and storage utilities could pose public health risks [34,43]. An ultraviolet (UV) lamp steriliser will be employed at the final stage of the rainwater treatment process to ensure the treated rainwater meeting microbiological terms of drinking water guidelines. For example, one of the validated UV systems offers a 99.99% reduction in bacteria, viruses and protozoan cysts that carries a powerful UV intensity of 40 $mJ/cm^2$ at a service flow rate up to 216 L/min with an instant contact time. Moreover, the UV lamp is low maintenance and easy to monitor via its weatherproof digital controller, indicating lamp performance, lamp fail alerts and lamp replacement indicator without interrupting water flow. Additionally, the

optional thermal relief valve ensures the UV lamp remains at optimal UV output temperature and protects the lamp from overheating [70].

*d.   Storage tank to hold daily produced drinking water*

The produced drinking water must meet drinking water quality standards, and is stored in a food-grade tank within a retention time of 24 h. The treated water should be distributed daily to points of use via a single tap or dispenser in each household. The selection of tank capacity depends on the daily community drinking water demand. The service of de-slugging and cleaning for this storage tank is considered every five years in the maintenance section of the proposed RWH system. Information about the type and availability of the drinking water storage tank can be found at local water tank industries in most nations [71].

*e.   Solar energy system and water pumps*

The use of solar energy can offer significant benefits to the RWH system by reducing the overall project's carbon footprint, removing electricity costs, and offering low maintenance, high performance and reliability. A 3 kW 48 V solar energy system is suggested to be sufficient for two pumps (2 × 740 W), multiple filters (3 × 300 W) and a UV lamp (172 W). For example, all-weather innovative solar panels, string inverters or enphase IQ micro-inverters can be applied to ensure optimal output yield of the solar energy system [72]. Additionally, jet pressure pumps are used to transfer water from the rainwater storage tank to the rainwater treatment system and household drinking water distribution pipelines. The drinking water distribution pipelines should be made from food-grade polymer materials supplied by certified plumbing industries [63].

### 3.3. Water Balance and Economic Analysis

The feasibility of the proposed RWH system was investigated in terms of reliability, life cycle cost and sensitivity analyses. The rainfall data of Australian and Vietnamese stations were considered individually by the model. The result of Australian RWH systems was processed by averaging the results of Springwood, Richmond and Bilpin stations, while the result of Vietnamese RWH system was the average results of TSH, NB and TDM stations.

### 3.3.1. Reliability Analysis

To simulate the performance of the proposed RWH system under different rainfall regimes, a water balance model (WBM) was developed in MATLAB. In this WBM, the behaviour of the proposed community-scale RWH system was simulated on a daily scale, where the principal inputs were roof area, daily rainfall and daily water demand and outputs were daily drinking water production by the RWH system. There are two modelling approaches in RWH: a yield before spillage (YBS) model, which generally provides a 10–15% overestimation of water production, compared with a yield after spillage (YAS) model [73]. To be conservative, the YAS model was adopted in this study. All the model inputs and outputs are shown in Figures 4 and 5.

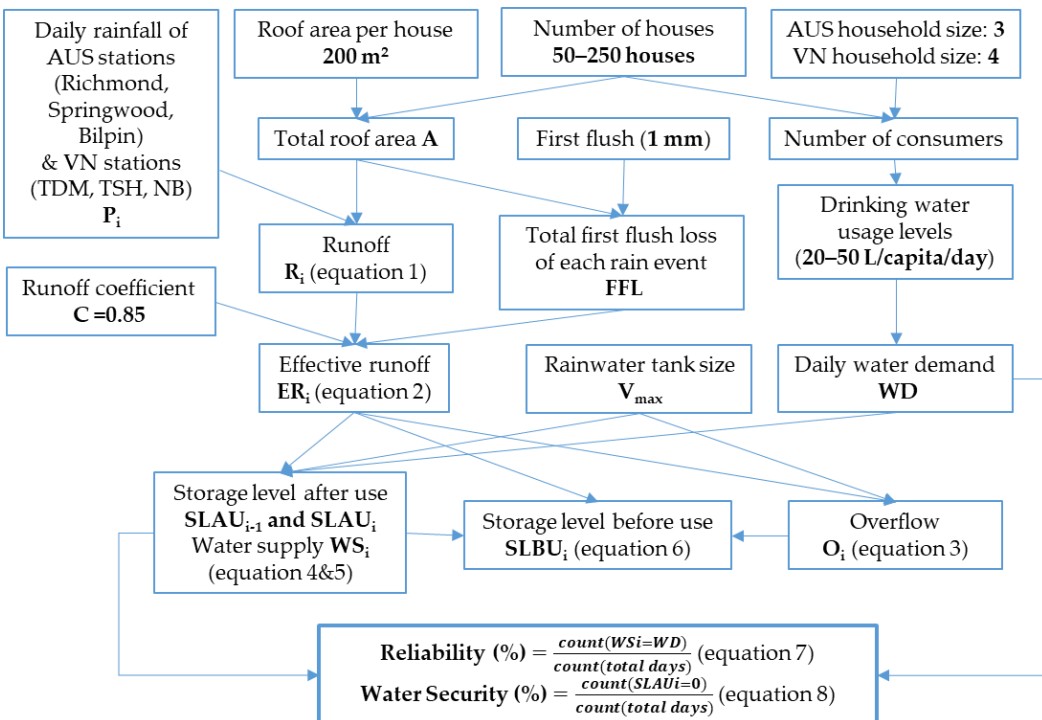

**Figure 4.** MATLAB modelling input and output parameters for reliability analysis.

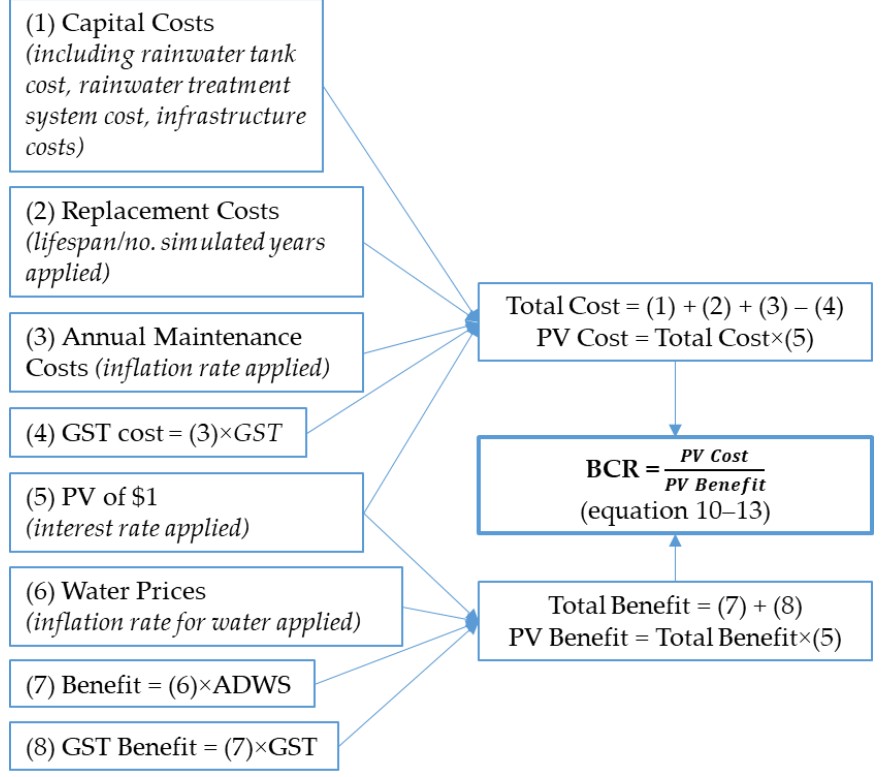

**Figure 5.** Procedure of BCR calculation used in the MATLAB model.

First, the utmost important parameter used in the model is the historical daily rainfall data of the selected rainfall station, which were imported to MATLAB as functions. Each function was incorporated into the model simulation in matrix format. MATLAB could recognise the daily rainfall values as well as the time series of rainfall values as tabular data in its Workspace. The size of each table includes 1 column and i rows; i = 21,915 for Australia and i = 14,610 for Vietnam.



Second, a number of inputs representing the analysis scenarios were entered intentionally by the model users including rural community sizes (the number of houses), household sizes, the daily drinking water usage required by a consumer and the rainwater tank size.

- Rural community sizes: due to different population sizes and densities between Australia and Vietnam, five scenarios of rural community with 50, 100, 150, 200 and 250 households were considered in the model.
- Household sizes: Vietnamese scenarios are based on household size of four people, rounded up from the average rural household size of 3.9 in the statistic of Vietnam Population and Housing Census survey [74]. However, three occupants per household is considered in Australian scenarios because households in Australia are getting smaller; the average number of people per household fell from 4.5 to 2.6 during 1911–2016 [75].
- Daily drinking water usage: In order to follow the UN requirements regarding the right to water, we selected drinking water demand in rural areas in the range of 20–50 L/capita/day (LCD). This range of drinking water demand for a person in a day can be sufficient for a range of activities (such as 10 L for drinking, 20 L for drinking and cooking, 30 L for drinking, cooking and personal washing, 40 L for drinking, cooking, personal washing and cloth washing, and 50 L for drinking, cooking, personal washing, cloth washing and cleaning home [76–78]). The study analysed multiple scenarios with 5 LCD intervals within the range of drinking water demand (i.e., 20, 25, 30, 35, 40, 45 and 50 LCD). As the study considered the rural areas with limited/no access to mains water supply, the analysis was performed to understand whether an RWH system could meet the drinking water demand for a rural community in the lowest expectation of emergencies and/or in the highest expectation of sustainable developments.
- Rainwater tank size: a wide range of tank sizes from 25–1600 kL was tested in the model to evaluate trends of reliability and the capacity of water supplied by the proposed RWH system.

Third, a number of constant values incorporated in runoff generation were adopted in the MATLAB model script including runoff coefficient, average roof area per house and average first flush loss per house.

- The runoff coefficient (C) is a dimensionless coefficient relating the amount of runoff to the amount of precipitation received. It depends on roof gradient and gutter characteristics. It is recommended that the C value for the roof is 0.75–0.95. The designers must determine the most appropriate C value within this range [79]; here we selected a typical value of 0.85.
- According to the guidance on the use of rainwater tanks by the Australian Government Department of Health, the average roof area can range from 100–150 m² for a small house, 150–200 m² for a medium house and greater than 200 m² for a large house [80]. In this study, the average roof area for a medium house in the rural community is considered at 200 m².
- First flush is the initial surface runoff from a rainfall event. There is considerable literature dedicated to the study of first flush phenomena. The classic study by Yaziz et al. (1989), with a number of experiments based on fixed volumes, described a rule-of-thumb of diverting 5 L of first flush [81]. Other publications have recommended first flush should be between 1–2 gallons per 100 square feet of roofing or 20–25 L for an average-sized roof [82]. Studies on quantifying the first flush phenomenon reported that for each 1 mm of first flush the contaminate load will halve. It is possible to remove up to 85% of incoming pollution material while retaining 85% of the roof harvested rainwater if the first flush device is designed carefully [83,84]. Moreover, other research showed that bypassing the first 2 mm of rainfall gives harvested rainwater the most quality parameters compliant with the Australian Drinking Water

Guidelines [85]. In this study, it is assumed that the rural community can apply well-designed first flush devices, and the first 1 mm of rainfall in a rain event can be the robust average first flush loss value that was applied in the MATLAB model.

- Finally, modelling outputs for the proposed RWH system were calculated as follows.
- Number of consumers = number of houses x number of occupants per household.
- Water demand (WD) = number of consumers x daily drinking water usage per person.
- Runoff, effective runoff, overflow, water demand, water supply and track changes in storage volume from the rainwater tank can be simulated on a daily time step by the YAS algorithm (Equations (1)–(7)).

$$R_i = P_i \times A/1000 \tag{1}$$

$R_i$ (kL) is runoff, $P_i$ (kL) is precipitation of day '$i$' and A (m²) is total roof area = 200 × number of houses.

$$If \ R_i \times C - FFL > 0, then \ ER_i = R_i \times C - FFL, else \ ER_i = 0 \tag{2}$$

$ER_i$ (kL) is effective runoff, C = 0.85 is runoff coefficient and FFL (kL) is total first flush loss for each rainfall event = A × 1 mm/1000.

$$If \ ER_i - V_{max} > 0, then \ O_i = ER_i - V_{max}, else \ O_i = 0 \tag{3}$$

$O_i$ (kL) is overflow and $V_{max}$ (kL) is the max volume of rainwater tank or tank size.

The YAS algorithm starts by setting the initial volume for the rainwater tank on the first day of model simulation.

$$\begin{aligned} If \ ER_1 - WD > 0, \ then \ WS_1 = WD, \ SLBU_1 = ER_1 - O_1 \ and \ SLAU_1 = SLBU_1 - WD \\ else \ ER_1 - WD < 0, then \ SLBU_1 = WS_1 = ER_1 \ and \ SLAU_1 = 0 \end{aligned} \tag{4}$$

$WD$ (kL) is daily drinking water demand, $WS$ (kL) is daily water supply, $SLBU$ (kL) is storage level before use and $SLAU$ (kL) is storage level after use.

From day 2 of model simulation, $SLAU$ is calculated by a complicated construction with relational and logical operators. First, if $SLAU$ of the previous day '$i$-1' combined with ER of day '$i$' could satisfy WD of day '$i$' and still remained less than tank size (*Vmax*), then SLAU of day '$i$' is the combination of SLAU of the previous day '$i$-1' and ER of day '$i$' after satisfying water demand of day '$i$' (Equation (5a)). Second, if SLAU of the previous day '$i$-1' combined with the ER of day '$i$' is less than WD of day '$i$', which means WS could not satisfy WD, then there is no SLAU due to that WS has taken all available water and left the tank empty (Equation (5b)). Third, if SLAU of the previous day '$i$-1' combined with ER of day '$i$' could satisfy WD of day '$i$' and is still greater than Vmax, then SLAU is Vmax, and WS could satisfy WD (Equation (5c)). SLBU of day '$i$' is SLAU of the previous day '$i$-1' combined with effective runoff after overflow of day '$i$' (Equation (6)). The procedure is repeated for each day of the year, and each year is dealt with separately. The annual WS, so-called ADWP, is averaged and used for LCCA of the proposed RWH system in each scenario.

$$If \ 0 \le SLAU_{i-1} + ER_i - WD < V_{max}, then \ SLAU_i = SLAU_{i-1} + ER_i - WD, \ WS_i = WD \tag{5a}$$

$$elseif \ SLAU_{i-1} + ER_i - WD < 0, then \ SLAU_i = 0 \ and \ WS_i = SLAU_{i-1} + ER_i \tag{5b}$$

$$else \ SLAU_{i-1} + ER_i - WD \ge V_{max}, then \ SLAU_i = V_{max} \ and \ WS_i = WD \tag{5c}$$

$$SLBU_i = SLAU_{i-1} + ER_i - O_i \tag{6}$$

The system reliability can be evaluated by the proportion of days when potable water demand is met by potable water produced by the proposed RWH system. On the other hand, the water security can be determined by the proportion of days when the proposed RWH system was empty (Equations (7) and (8)).

$$\text{Reliability (\%)} = \frac{Total\ no.\ of\ days\ when\ water\ demand\ is\ met\ by\ water\ supply}{Total\ no.\ of\ simulated\ days} \times 100 \tag{7}$$

$$\text{Water Security (\%)} = \frac{Total\ no.\ of\ days\ when\ storage\ tank\ is\ empty}{Total\ no.\ of\ simulated\ days} \times 100 \tag{8}$$

The reliability percentage was calculated by the proportion between the number of days that the drinking water produced/supplied by the RWH system could meet the drinking water demand and the total number of simulated days. The amount of produced drinking water that satisfied the drinking water demand on the daily time steps in the simulated period was known as the daily drinking water production of the RWH system, and hence, ADWP is the average of water supply on the yearly time steps in the simulated period.

3.3.2. Life Cycle Cost Analysis (LCCA)

This study used the LCCA to assess costs and benefits of the proposed RWH system over its project lifespan in monetary terms following the AS/NZ Standard AS4536 "Life Cycle Costing—an Application Guide" [86]. The financial benefit was generated from converting the drinking water production into monetary value using selected water price. Alternatively, the life cycle costs of the system were analysed by financial considerations of capital costs, replacement costs and maintenance costs. All past, present and future cash flows were converted to present dollar value and are a function of discount rates. The concept of nominal cost (e.g., estimated changes in prices, efficiency, inflation, deflation and technology) and nominal discount rate to convert nominal cost to discounted cost were applied in the LCCA (Equations (9)–(12)). The procedure of LCCA was carried out by the MATLAB model, which incorporated Australian and Vietnamese economic data and the results from the reliability analysis (Figure 5).

$$C_D = C_N \times \frac{1}{(1+d_n)^y} \tag{9}$$

$C_D$ is discounted cost, $C_N$ is nominal cost, $d_n$ is nominal discount rate per annum and y is project lifespan in years.

$$\text{Discount rate} = \frac{1}{(1+i)^t} \quad \text{PV} = \frac{CF}{(1+i)^t} \tag{10}$$

PV is present value, CF is the cash flow, *i* is the interest rate and *t* is the year in which the cash flow occurred.

$$\text{NPV}(i,y) = \sum_{t=0}^{y} \frac{CF_t}{(1+i)^t} \tag{11}$$

NPV is net present value which is the sum of PV over the lifespan; CF is the difference between cash outflow and inflow reduced by the discount rate appropriate to the time (*t*) of transaction.

$$\text{BCR} = \frac{\sum_{t=0}^{y} \frac{B_t}{(1+i)^t}}{\sum_{t=0}^{y} \frac{C_t}{(1+i)^t}} \tag{12}$$

BCR is the ratio of the benefit over the cost of the time (*t*) of transaction.

In relation to the capital costs of the proposed RWH system (Table S1), an assumption of the rainwater treatment system cost was made with the highest daily water production scenario of 50 kL/day for 250 houses (1000 consumers) at the high demand level of 50 LCD. It should be noted that this assumption is to simplify the capital cost for the rainwater treatment system. The tank containing treated water for daily water demand was considered at 20 kL. This one-size-fits-all approach became the uniform consideration for

other scenarios regardless of lower daily water demand. Capital, replacement and maintenance costs of all items related to the rainwater treatment system section would remain fixed. In fact, the cost savings from a smaller tank less than 20 kL would be negligible compared to the chosen fixed cost. In addition, it would be less cost effective to custom-make smaller items for the rainwater treatment system. As for the replacement costs of the proposed RWH system (Table S2), replacement items on an annual basis include UV lamp, calcite media and sediment filter membrane. The frequency of other replacements will depend on product warranties. Additionally, concerning the maintenance cost of the proposed RWH system (Table S3), a five-year service interval is mandatory to comply with the Kingspan pro rata product warranty that ensures the rainwater tank is performing optimally. Similarly, the storage tank for treated drinking water must be de-slugged every five years. The rainwater treatment system and cleaning roof catchments are suggested to be maintained on an annual basis by Kingspan Bronze Package, including replenishing calcite media and cleaning/replacing the sediment filter, quartz sleeves, UV sensor, etc.

The general economic situation in a country would affect the LCCA of an RWH system because product costs increase with the inflation rate, while the present value of the dollar decreases annually by interest rate. Low inflation rate and high interest rate are more favourable to an RWH system [87]. Other research used the economic data of the developed country to apply to the developing one when comparing the LCCA of their RWH systems, which was influenced by various considerations and situations [88]. However, LCCA should be based on the local economic data of the country where the RWH system is located.

For Australia, the interest rate has decreased from 4 to 2%, while the inflation rate has been fluctuating between 0.87–3.33% in the last decade [89,90]. The Independent Pricing and Regulatory Tribunal (IPART) estimated expected inflation of 2.3% in 2020–2025, but Sydney Water argued that a lower inflation expectation of 1.7% should be adopted for their water pricing. The selected mains water cost from Sydney Water at non-drought drinking water usage charges followed the price scheme established by IPART for 2020–2024 [91,92]. The Australian bottled water price can vary from 0.40–4.38 AUD/L, including design, manufacturing and logistics expenses. It is complex to account for all variables involved in these constituent costs, and hence, the lowest bottled water price was considered in the Australian economic parameters.

For Vietnam, the interest rate was flat, about 6%, while the inflation rate fluctuated between 0–7.5% in 2013–2020 [93,94]. However, both rates are found to be dropped at around 4% due to the impact of the present global pandemic with its economic fallout [95,96]. Water tariff in HCMC is assigned with household water consumption levels; for example, the water price in 2019–2022 was set at 14,400 VND/m$^3$ (or 0.85 AUD/kL) for households with 6 m$^3$/person/month [97]. It is often more expensive to purchase bottled water of smaller capacity (e.g., 500 mL or 1.5 L) rather than large capacity (20 L), and thus, the price of 20 L bottled water was considered for the Vietnamese economic parameters.

In this study, electricity cost was not accounted for due to planned solar energy generation. Moreover, water costs were considered in a range from the mains water price at the government policy rate to the bottled water price at the commonly used rate. In the case of the top-up requirement for an empty tank in times of drought or emergency, the cost of bulk water from a truck tanker should be considered as one of the water costs as well. A summary of economic parameters used in the LCCA is given (Table S4).

In summary, the feasibility analysis of the proposed RWH system required a number of water demand scenarios. In total, each country has 35 scenarios, which were defined based on 7 drinking water usage levels (20, 25, 30, 35, 40, 45 and 50 LCD) paired with 5 community sizes (50, 100, 150, 200 and 250 houses). The modelling was performed to evaluate the effects of variable inputs on the outputs of reliability percentage, ADWP and BCR (Figures 4 and 5). The analysis covered both favourable and unfavourable outcomes

that may arise in the practical implementation of the Australian and Vietnamese RWH systems.

## 4. Results

### 4.1. Reliability

It should be noted that in each drinking water usage level, the daily water demand scenarios were increased proportionally by the number of houses chosen with a fixed interval (50 houses). The daily water demand would have a direct effect on the rainwater tank size selection and the system reliability of the considered scenarios. Varied tank sizes were tested individually by the model to provide the reliability and ADWP trends. There are several reasons influencing the differences in the rainwater tank size ranges for the Australian and Vietnamese systems. Firstly, although both systems considered the same values of drinking water usage levels and the same number of houses in each comparable scenario, the Vietnamese system had a higher number of consumers due to Vietnamese household size being four, compared to three occupants per Australian household (Section 3.3.1 Reliability Analysis). As a consequence, the daily drinking water demand values of the Vietnamese system were always higher than that of the Australian system, about 1.33 times (4:3). Moreover, average annual rainfall in Vietnam between 1980–2019 was about 1.74 times higher than the average annual rainfall in Australia between 1960–2019 (1844 mm:1059 mm) (Table 1). The dry season for Vietnam lasts for four months (Dec–Mar), with an average rainfall of about 19 mm/month. In case of selected stations in Australia in NSW, the dry season lasts for three months (Jul–Sep) with an average rainfall of 49 mm/months. To meet the water demand in dry seasons, the rainwater tank size needs to be bigger for Vietnam as compared with Australia to reserve a higher volume of water to achieve a similar level of reliability. As a result, the Vietnamese tank size range was about 3.43 times higher than that of Australia. This approximate ratio can be used to predict an optimised tank size (OTS) for each scenario. For example, at 50 LCD and 750 users (equals to 37.5 kL/day water demand), the Australian OTS was 1400 kL, while the Vietnamese OTS was 4800 kL for 50 LCD and 1000 users (equal to 50 kL/day water demand) (Figure 6). Another point worth noting is that the reliability percentage improved with a larger tank that is out of range or beyond these optimised values, but provided no further improvement upon reaching the equilibrium of reliability (≥99%), as found in other studies [14,25,98] (Figure 6).

It is found that both the systems could achieve a similar reliability level with a proportional series of rainwater tanks (Figure 6). According to the correlation between daily drinking water demand and OTS, the Australian and Vietnamese RWH systems could meet the fixed reliability of 90% with their own basis of storage volume, "a unit of OTS per unit of daily water demand". In particular, each kL of daily drinking water demand would require around 28 kL of the Australian rainwater tank volume and about 84 kL of the Vietnamese tank storage (Figures 6 and 7). Based on this observation, the correlation between OTS and daily drinking water demand of the Australian and Vietnamese RWH systems can be established via the linear regression equations $y = 0.03x + 2.59$ ($R^2 = 0.96$) and $y = 0.01x + 1.27$ ($R^2 = 0.99$), respectively (Figure 7).

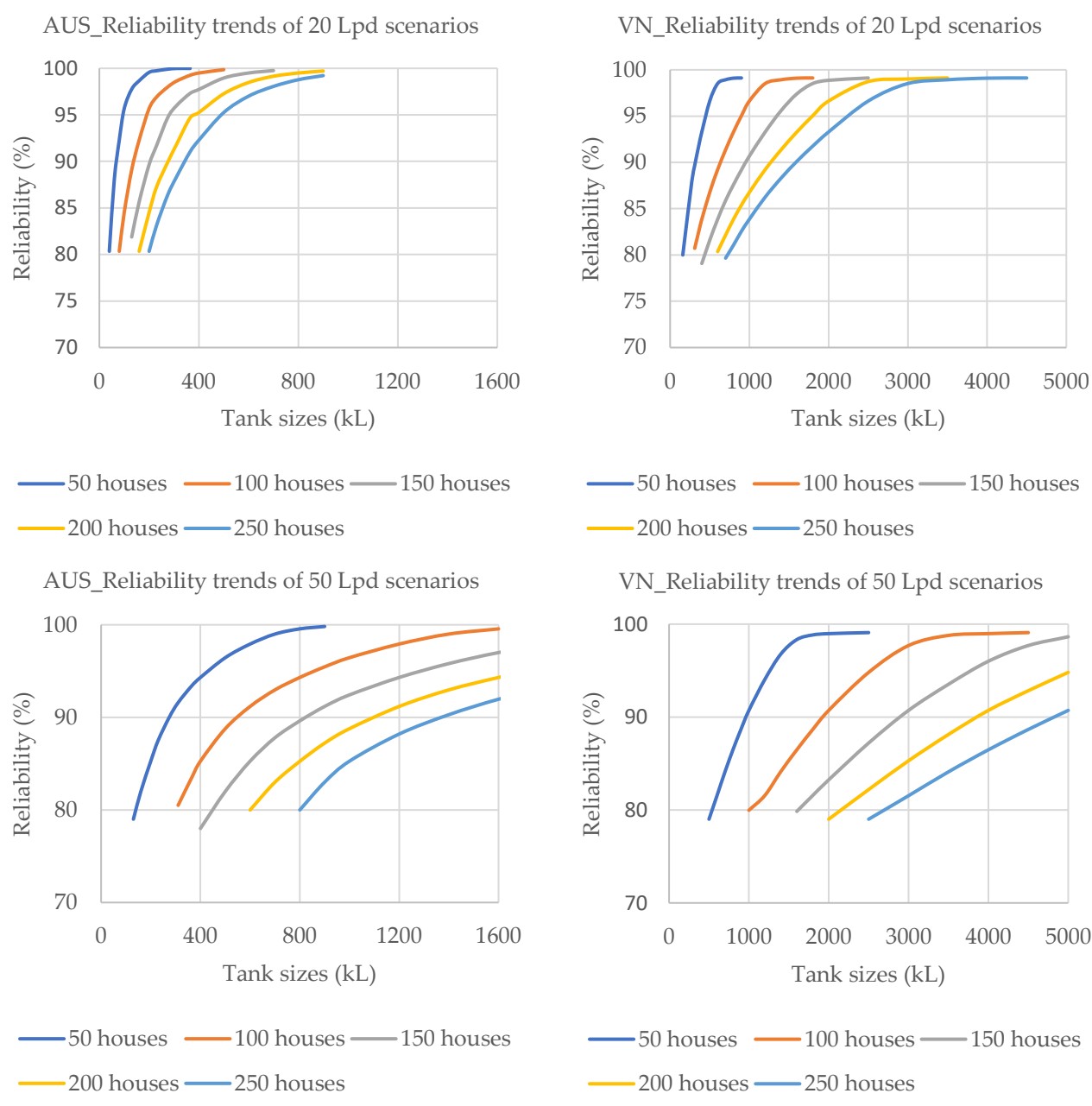

**Figure 6.** Reliability trends with respect to tank sizes and scenarios for Australian and Vietnamese RWH systems.

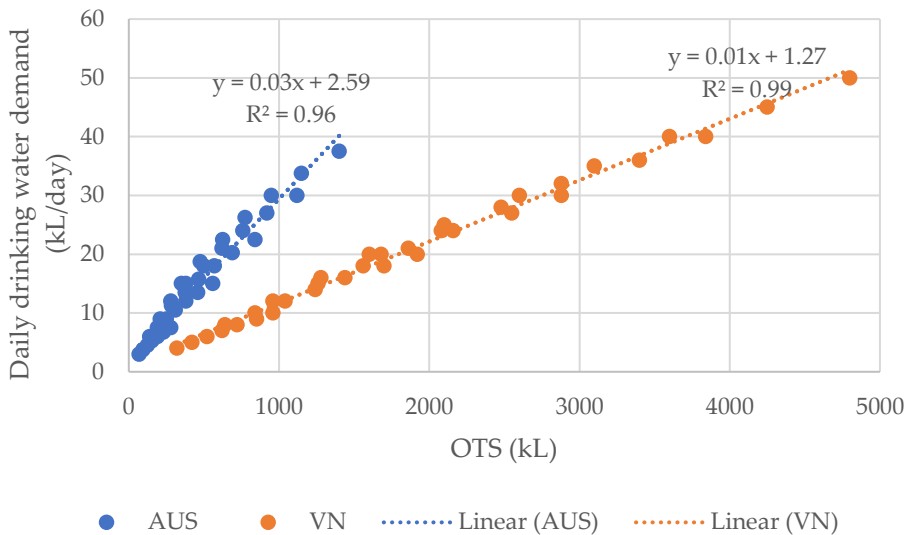

**Figure 7.** Correlation between daily drinking water demand and optimised tank size for the Australian and Vietnamese RWH systems.

*4.2. ADWP*

The results of ADWP also had proportional patterns with respect to tank sizes and scenarios because the results of reliability and ADWP shared the same input parameters in the reliability analysis (Figure 8). It can be emphasised that the input of Australian or Vietnamese rainfall data only had an effect on the difference between Australian and Vietnamese tank size ranges, as mentioned earlier in Section 4.1, which also aligned with the ADWP trends with respect to tank size (Figure 8). Moreover, the ADWP of both systems were increased not only by increasing tank size but also by increasing drinking water demand (Figure 9). Therefore, it would be more informative to observe the ADWP patterns with OTS or daily drinking water demand separately. As can be seen, both systems showed very similar correlations between daily drinking water demand (x) and ADWP (y), which can be demonstrated by the average linear regression equation $y = 329.39x + 16.84$ ($R^2 = 1.00$) (Figure 9). Moreover, the correlation between ADWP and OTS of the Australian and Vietnamese systems can be shown via the linear regressions: $y = 8.89x + 858.18$ ($R^2 = 0.96$) and $y = 3.43x + 428.04$ ($R^2 = 0.99$), respectively (Figure 10).

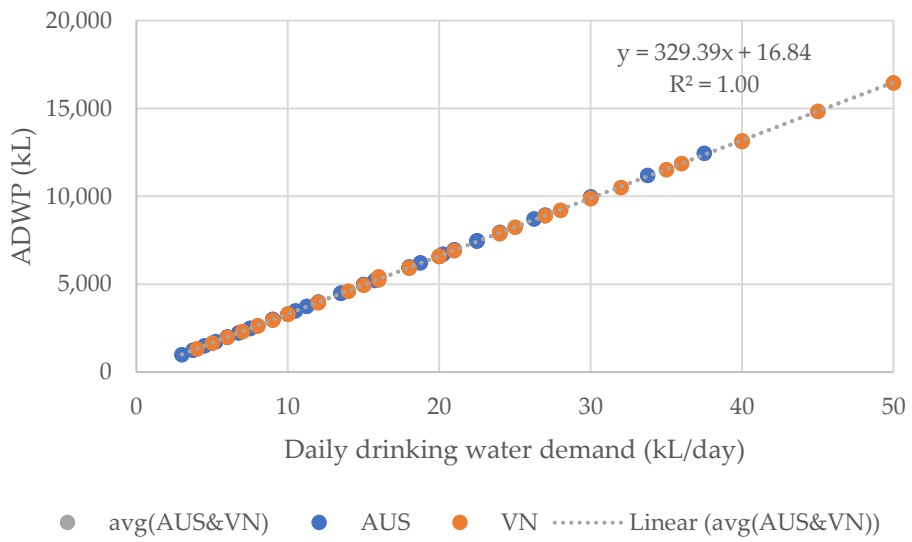

**Figure 8.** ADWP trends with respect to tank sizes and scenarios for Australian and Vietnamese RWH systems.

**Figure 9.** Correlation between daily drinking water demand and ADWP for the Australian and Vietnamese RWH systems.

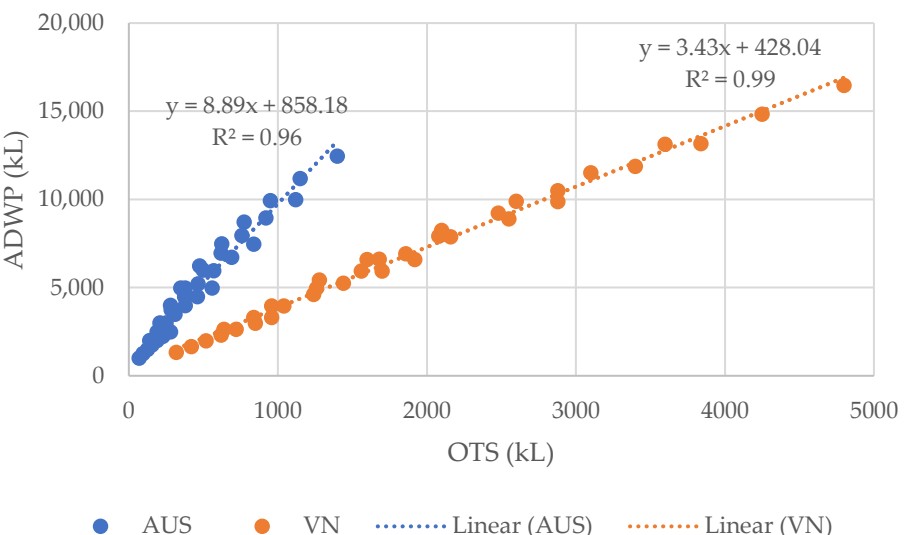

**Figure 10.** Correlation between ADWP and OTS for the Australian and Vietnamese RWH systems.

*4.3. BCR*

The LCCA was carried out using the results of OTS and ADWP with further model inputs including the number of simulated years, the selection of water prices and economic parameters. The BCR results with respect to tank sizes and drinking water usage scenarios would strongly clarify the relationship between costs (depending on the selection of OTS) and benefits (based on ADWP) at comparable economic situations that applied for drinking water supply in Australia and Vietnam (Figures 11 and 12). On initial inspection, both Australian and Vietnamese systems could not achieve a break-even point of BCR = 1 in any scenario because both the Australian and Vietnamese mains water prices were too low to be the converted monetary values for the LCCA (Figure 11). Moreover, the variations in the BCR between the drinking water usage levels (20–50 LCD with 5 LCD intervals) were about 0.1 for Australia systems and 0.05 for Vietnamese systems (Figure 11).

As the LCCA with the mains water price could not provide the desired financial feasibility (i.e., BCR ≥ 1.00) for both the systems, further LCCA was carried out to ascertain the BCR patterns at truck tanker and bottled water prices (Figure 11). It can be seen that by using OTS, both the systems could reach the reliability of 90% and achieve the same ADWP level in each scenario (Figure 9), which means that they both gained similar economic outcomes of the ADWP. The higher scenario with the higher daily drinking water demand would result in the higher BCR values owing to the greater ADWP as well as the converted monetary value of the ADWP (Figure 11). However, the Vietnamese system would require rainwater tank storage around 3.43 times larger than that of Australia (Figure 7). The Vietnamese system cost would be accordingly higher than the Australian one. The bigger tank and system costs would reduce the BCR values of the Vietnamese system. As a result, the BCR values of the Australian system were always higher than those of the Vietnamese system (Figure 11). Finally, the use of truck tanker drinking water price or bottled drinking water price resulted in unpractical BCR values that were far exceeding the desired threshold of BCR = 1.

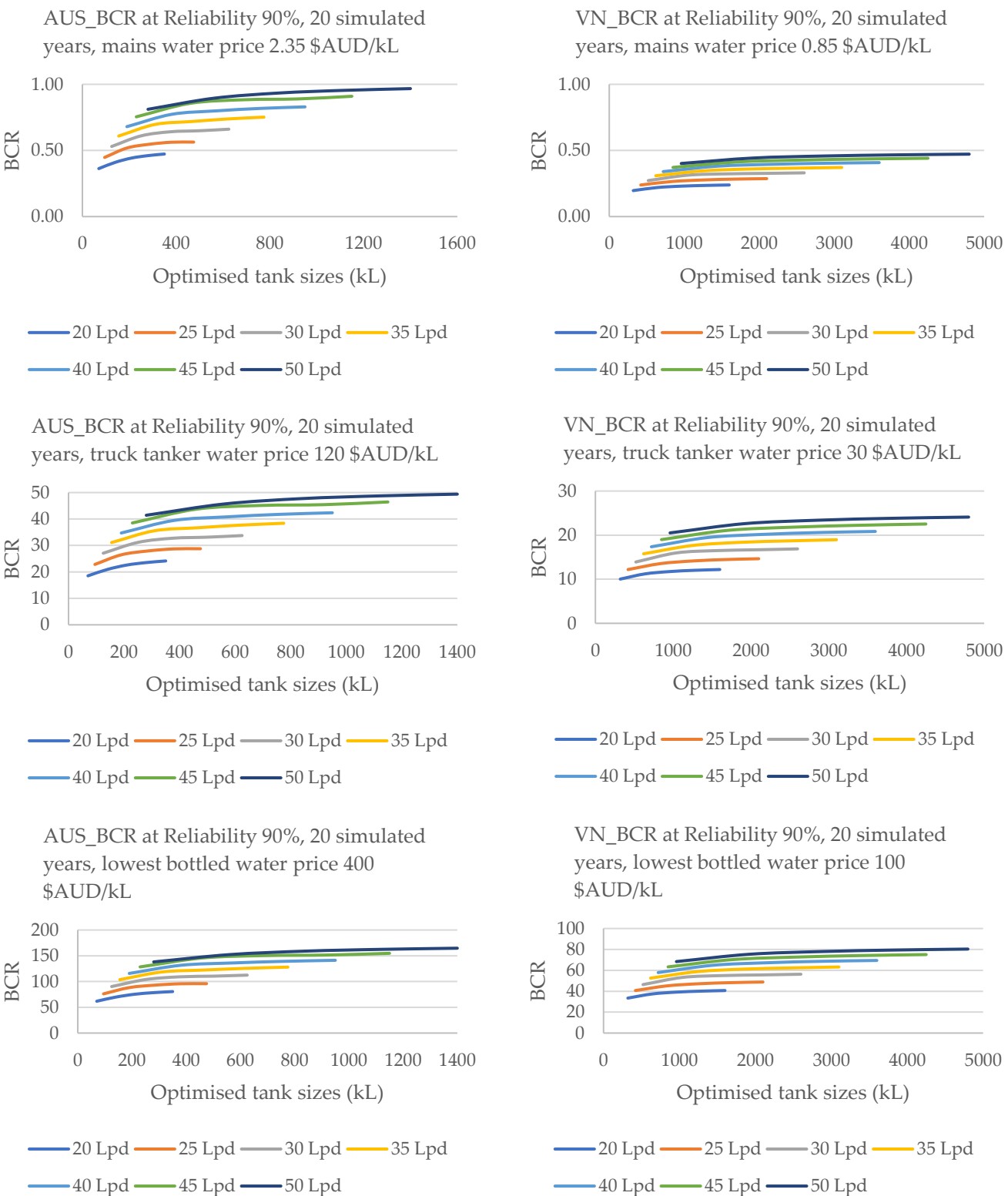

**Figure 11.** BCR values for the Australian and Vietnamese RWH systems considering reliability 90%, 20-year lifespan at mains, truck tanker and bottled drinking water prices.

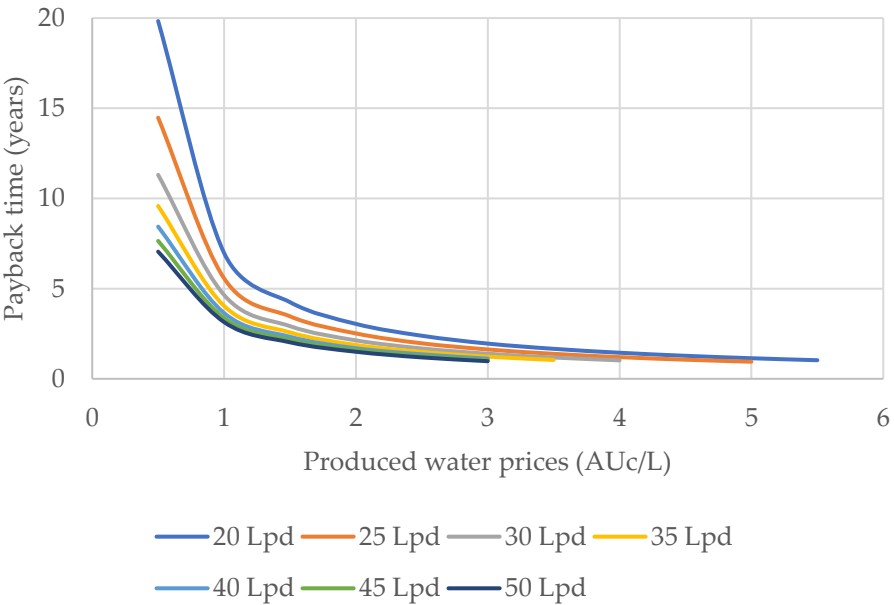

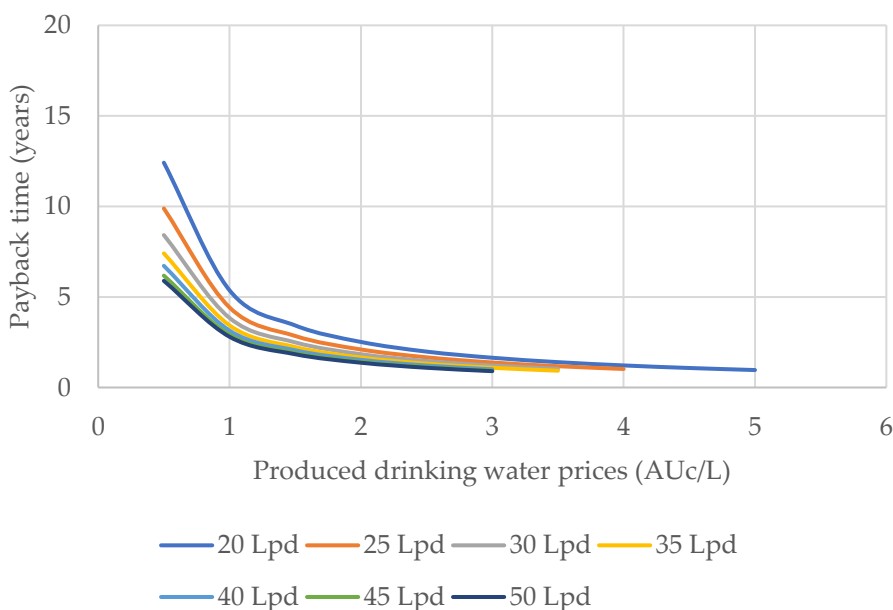

**Figure 12.** Correlation between produced water prices and payback time for the Australian and Vietnamese RWH systems with respect to the highest number of users and scenarios of drinking water usage.

*4.4. Sensitivity Analysis for the Produced Water Price*

The sensitivity analysis was carried out to estimate a reasonable and affordable range of produced drinking water prices, which should fall in between the mains water price and the bottled water price. The correlation between the produced water prices and the payback time are similar for both the systems. In general, payback time could be reduced by increasing the produced water price. For instance, in the highest scenario of 50 LCD, 250 houses, 750 users and OTS 1400 kL, the Australian system could reduce the payback period by three times (from 3.1 to 1.0 years) if the produced water price was increased by

three times (from 0.01 to 0.03 AUD/L). Likewise, in the highest scenario of 50 LCD, 250 houses, 1000 users and OTS 4800 kL, if the produced water price was increased by three times (from 0.01 to 0.03 AUD/L), the Vietnamese system could reduce the payback period by three times (from 2.8 to 0.9 years) (Figure 12). In addition, the results of sensitivity analysis showed that by applying the considered affordable price of 0.01 AUD/L for the produced drinking water, both the Australian and Vietnamese systems could achieve financial return within 7.0–3.1 years and 5.4–2.8 years, respectively (Figure 12). This proved to be a good sign for project investors due to that the systems can be paid back within a timeframe that is shorter than the project lifespan. In the case of when the drinking water usage level was dropped to the lowest, 20 LCD, in order to ensure the system could be paid back in 20-year lifespan, the produced water prices need to be reduced to 0.50 AUD/L (Figure 12).

The minimum price of the produced drinking water is the most important parameter to consider in designing the proposed RWH system, which can be at the affordable low price of 0.01 AUD/L. If the produced drinking water price was fixed at 0.01 AUD/L, the payback time would only be influenced by the water demand, which was based on the number of houses/consumers in the community and the drinking water usage levels. All Vietnamese scenarios with drinking water demands between 4–50 kL/day were feasible at the considered price and lifespans (Figure 12). However, a number of Australian scenarios with a small number of houses/consumers and small daily drinking water demands (below 6 kL/day) were not fit-for-purpose and cannot be implemented, due to their payback period exceeding the 20-year lifespan, unless the produced water price was increased beyond 0.01 AUD/L (Figure 13).

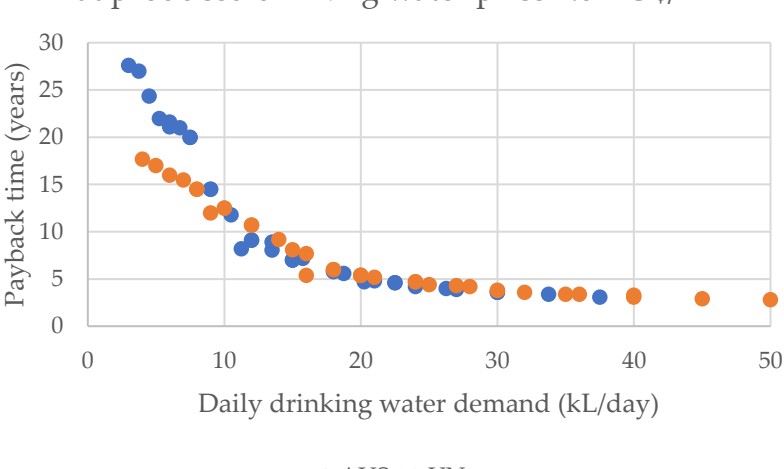

**Figure 13.** Payback time with respect to the produced drinking water price 0.01 AUD/L and daily drinking water demand for Australian and Vietnamese RWH systems.

## 5. Discussion

The selection of OTS by factoring in the minimum cost criteria, in which the economic convenience of large tanks decreases as rainwater availability decreases, is a commonly used approach by many previous studies [14,29]. The selection of OTS was based on parameters influencing the size of the rainwater tank, such as local rainfall, daily drinking water demand levels and the number of consumers [42]. This study used the approach of selecting OTS to provide at least 90% reliability and the optimum ADWP for the Australian and Vietnamese RWH systems in 70 scenarios, considering multiple drinking water usage levels and community sizes. Moreover, this study applied different rainfall

regimes in Australia and Vietnam to show the differences in tank size ranges as well as OTS between the two countries (Figure 7).

An important point is that drinking water is essential for any rural community to survive all year round. As such, it is vital to prioritise the RWH system with a minimum 90% reliability by using the OTS strategy. A shortage of drinking water supply by about 10% can be tolerated, as people can purchase bottled water if needed in the insecure time. The trends of reliability clear show that the system reliability would improve between 95–99% with tank sizes larger than OTS (Figure 6). This improvement for reliability provided no further improvement for the BCR, which has been proven in another reliability analysis for a small-scale RWH system [73]. In addition, it will not be practical for the RWH system to go beyond the optimised tank size range because the additional costs of oversized tanks will outweigh the equilibrium benefit of ADWP.

Past research has reported that water savings of a single-household RWH system for non-potable uses would substantially depend on the number of occupants per household. However, if the number of users was unchanged, the water savings would not increase with a larger tank because a significant portion of harvested water would remain unused [13,14,73]. In other studies, the "water savings" term is equivalent to the "drinking water production" term used here. According to this principle, this study applied multiple scenarios of drinking water usage levels (20–50 LCD) coupled with 150–1000 consumers to establish a comprehensive correlation of water demand and water supply for the Australian and Vietnamese RWH systems (Figures 8 and 9). In addition, the relationship between ADWP and corresponding OTS were found to be useful for the LCCA of the Australian and Vietnamese RWH system (Figure 10).

The role of OTS is very important to both the reliability assessment as well as the BCR estimation. In fact, the BCR was influenced directly by ADWP and corresponding OTS. Both the Australian and Vietnamese systems may achieve similar economic values of ADWP in comparable scenarios. However, the Vietnamese system would require larger tank storage, which resulted in higher costs and lower BCR for the Vietnamese system compared to the Australian system (Figure 11). Additionally, this study applied bottled water and truck tanker water prices to provide insights into the potential BCR values associated with the market prices for drinking water (Figure 12), as found in other research [25]. Australian water prices are always higher than Vietnamese ones in each type of water price, which is another key factor that means the BCR of the Vietnamese system could never compete with that of Australia.

Within the 20-year simulation, the break-even point of BCR = 1 could not be achieved at mains drinking water prices. Although the systems might reach the BCR break-even point with a greater number of simulated years, it would involve a number of further parameters with their own inherent uncertainties and complications, including product warranties, system replacements and maintenance services. It is suggested that rainwater tank rebates should be applied to minimise these undefined factors. Therefore, government rebates are necessary for both the Australian and Vietnamese systems to support their financial viability. It is believed that if the proposed RWH system is implemented, business decision-makers and policymakers must prioritise the system reliability rather than its financial affordability. This finding is in agreement with Australian government rebate policies about RWH, which mentioned that a rebate would cover the tank cost, making the larger tank the most viable and help to supply or produce more water [99]. This finding may also encourage the Vietnamese government to consider a rebate to help implementation of RWH systems for rural communities.

None of the prices of mains water, truck tanker water and bottled water were able to prove the financial viability of the RWH system and could not become the standardised unit prices of the produced drinking water. This study has shown clearly that both the Australia and Vietnam scenarios achieve similar correlations between payback time and produced water prices, and therefore, the produced drinking water price of the proposed RWH system within the 20-year lifespan can be standardised at 0.01 AUD/L. This is in line

with other studies, for example, the operation and maintenance cost of the RWH system with onsite drinking water treatment for 100–500 people was found to be 0.03 USD/L (about 0.04 AUD/L) [26].

There are several similar investigations on large-scale RWH systems around the world [24,29,100], but they rarely considered advanced system designs for drinking purposes, such as rainwater treatment technologies associated with other aspects of drinking water supply for rural communities. Thus, the proposed RWH system in this study is novel due to its inclusive community characteristics and comprehensive system design. The exponential increase in the global population and rising living standards are likely to place extreme pressure on global development; thus, the proposed RWH system is essential to foster this growth in a sustainable manner.

It should be noted that rainwater may need some simple treatments to avoid waterborne diseases among the consumers in rural areas. In this regard, Huang (2021) [101] and Alim et al. (2020) [25] presented simplified treatment techniques to treat harvested rainwater. Furthermore, Kearns and Flanagan (2007) and Kearns (2016) proposed affordable charcoal filtration and biochar adsorbents to treat rainwater, respectively [51,102]. In addition, Kearns et al. (2021) presented a combined method of biochar with UV to remove organic micropollutants from water [52]. These show that simplified treatment methods can be used to enhance the quality of harvested rainwater.

## 6. Conclusions

This study presents the viability of community-scale rural RWH systems for the selected areas in Australia and Vietnam. It was found that Vietnam needs a bigger tank size than Australia to achieve the same supply reliability, which led to a reduction in the BCR for Vietnam. This is due to the fact that Vietnam has a larger family size and more intense dry season. This study shows the price of the produced drinking water from the proposed RWH system is 0.01 AUD/L, and Vietnam can adopt it with daily drinking water demand of 4–50 kL/day, 200–1000 users and 20–50 LCD, while Australia can only adopt it with daily drinking water demand >6 kL/day, 150–750 users and 20–50 LCD. It was also found that the current mains water price is too low to make the RWH system financially viable.

Since Australia has quite different spatial rainfall characteristics, the outcomes of this study (based on three selected stations in NSW) should not be generalised for the whole of Australia; however, the developed tools can be applied to any part of Australia to reflect the local rainfall condition into the modelling output. The above remark is applicable to Vietnam as well. Indeed, the developed tools can be applied to any country, where the outcomes need to be interpreted in the context of the selected area.

Further investigation on the aspects of rainwater treatment technologies, legal costs and approval processes are recommended. Different types of water disinfectants (e.g., chlorine, chloramine and ozone) can be used as a substitute for UV technology, as they may be unaffordable and/or unavailable in many developing countries. With these solutions, we intend to provide a pathway to reducing the global shortage of quality drinking water in rural communities, which will contribute towards achieving the UN's water-related Sustainable Development Goals.

**Supplementary Materials:** The following supporting information can be downloaded at: www.mdpi.com/article/10.3390/w14111763/s1, Table S1: Capital costs of the proposed RWH system; Table S2: Replacement costs of the proposed RWH system; Table S3: Maintenance costs of the proposed RWH system; Table S4: Economic parameters for use in the LCCA.

**Author Contributions:** T.T.R. conceptualised the study, developed the water balance model, gathered data, carried out all the analyses and drafted the manuscript. M.A.A. closely supervised the analysis, assisted in the interpretation of results and revised the draft. A.R. supervised the analysis, assisted in the conceptualisation of the problem, checked the results and revised the draft. All authors have read and agreed to the published version of the manuscript.



**Funding:** No specific funding was received for this study from the private or public sector.

**Data Availability Statement:** The data used in this study can be obtained from the Australian Bureau of Meteorology and Vietnamese Southern Regional Hydro-meteorological Centre by paying a prescribed fee.

**Acknowledgments:** The authors ensure that all individuals included in this section have consented to the acknowledgement. The authors acknowledge the traditional custodians of the land upon which the study was based. We would like to thank the Australian Bureau of Meteorology and Vietnamese Southern Regional Hydro-meteorological Centre for providing rainfall data and local suppliers to for providing data on the cost of RWH system components. The authors would like to thank Giles M. Ross (Researcher, Hawkesbury Institute for the Environment, Western Sydney University) for his time proofreading this manuscript, and Aman Khatri (Computer Engineer, Tennessee USA) for consulting the MATLAB modelling script. The authors also acknowledge four anonymous reviewers who provided constructive suggestions, which have improved the quality of the manuscript.

**Conflicts of Interest:** Authors declare that there is no conflict of interest.

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
