# Peer review of "Community-Scale Rural Drinking Water Supply Systems Based on Harvested Rainwater: A Case Study of Australia and Vietnam"

_water, doi:10.3390/w14111763_

Round 1

Reviewer 1 Report

paper has significantly improved after your revisions, congratulations for the interesting work 

Author Response

Thanks for accepting the revised paper.

Reviewer 2 Report

The paoer is well written and ready for publication 

Author Response

Thanks for accepting the revised manuscript.

Reviewer 3 Report

Lines 32-98, 1. Introduction

The literature can be more comprehensive. You can benefit from the following reference to internationalize the literature:

Anabtawi, F., Mahmoud, N., Al-Khatib, I.A., Hung, Y.-T. Heavy Metals in Harvested Rainwater Used for Domestic Purposes in Rural Areas: Yatta Area, Palestine as a Case Study. Int. J. Environ. Res. Public Health 2022, 19, 2683. https://doi.org/10.3390/ ijerph19052683

Line 237, “d. Storage tank to hold daily produced drinking water”. Something should be written about the frequency of cleaning and disinfection of the tank.

Lines 293-295, a reference should be added to the paragraph.

Author Response

Thanks for your constructive suggestions, which have been considered as noted below in preparing the revised manuscript.

  1. Lines 32-98, 1. Introduction: The literature can be more comprehensive. You can benefit from the following reference to internationalize the literature: Anabtawi, F., Mahmoud, N., Al-Khatib, I.A., Hung, Y.-T. Heavy Metals in Harvested Rainwater Used for Domestic Purposes in Rural Areas: Yatta Area, Palestine as a Case Study. Int. J. Environ. Res. Public Health 2022, 19, 2683. https://doi.org/10.3390/ ijerph19052683

 Response: Thank you for your useful comment. We have used the suggested reference [reference no. 30] together with about 20 new references in the revised literature in the introduction section.

  1. Line 237, “d. Storage tank to hold daily produced drinking water”. Something should be written about the frequency of cleaning and disinfection of the tank.

 Response: We appreciated your comment.  We have added Lines 288-290 to address the comment.

  1. Lines 293-295, a reference should be added to the paragraph.

Response: Thank you for your suggestion. We have rewritten the paragraph (Lines 346-354) with 2 supporting references [59] and [60].

Reviewer 4 Report

This research proposes the design of community-scale supply systems of drinking water based on harvested rainwater for rural Australia and Vietnam. In general, the technical calculation and test are pretty compelling. However, there are critical issues that require significant revisions/additions. The following comments intend to enable the authors to disseminate their work at the highest possible quality.

LITERATURE REVIEW.

  • This article misses a "Literature Review" section. After the background story of this research (Introduction), this research requires a rigorous review of the literature to establish the framework of the intended water supply system. The authors should review the literature relevant to crucial issues, including but not limited to water supply systems, rainwater harvesting, the characteristics of drinking water, and community-scale technological solutions.
  • The review should gather components necessary to build the intended supply system of drinking water based on harvested rainwater. The components would be the basis to form the underlying structure of the technical design of the proposed water supply system. Without a rigorous literature review on the necessary components, the structural design of the proposed water supply system is arbitrary and scientifically baseless.

ENGINEERING DESIGN.

  • This article misses an "Engineering Design" to guide the design process of the proposed water supply system. Engineering design is fundamental for any technology design, by which every part/sub-system of the technology is methodically designed and not based on the designers' arbitrary decisions.
  • The authors can choose from existing engineering design methods for general purposes (e.g., Pahl & Beitz's; Dym & Little's; Dieter & Schmidt's Engineering Design) or rural-specific purposes (e.g., Sianipar et al.'s Design Methodology for Appropriate Technology; Pearce's Open-Source Appropriate Technology).
  • By following a formal engineering design method, the authors can argue how they methodically identify and solve precise problems through part-by-part of the system's structure. The performance of the proposed water supply system might be testable empirically; however, without a formal engineering design, the proposed design of the water supply system cannot be proven scientifically.

Author Response

This research proposes the design of community-scale supply systems of drinking water based on harvested rainwater for rural Australia and Vietnam. In general, the technical calculation and test are compelling. However, there are critical issues that require significant revisions/additions. The following comments intend to enable the authors to disseminate their work at the highest possible quality.

  1. LITERATURE REVIEW.
  • This article misses a "Literature Review" section. After the background story of this research (Introduction), this research requires a rigorous review of the literature to establish the framework of the intended water supply system. The authors should review the literature relevant to crucial issues, including but not limited to water supply systems, rainwater harvesting, the characteristics of drinking water, and community-scale technological solutions.
  • The review should gather components necessary to build the intended supply system of drinking water based on harvested rainwater. The components would be the basis to form the underlying structure of the technical design of the proposed water supply system. Without a rigorous literature review on the necessary components, the structural design of the proposed water supply system is arbitrary and scientifically baseless.
  1. ENGINEERING DESIGN.
  • This article misses an "Engineering Design" to guide the design process of the proposed water supply system. Engineering design is fundamental for any technology design, by which every part/sub-system of the technology is methodically designed and not based on the designers' arbitrary decisions.
  • The authors can choose from existing engineering design methods for general purposes (e.g., Pahl & Beitz's; Dym & Little's; Dieter & Schmidt's Engineering Design) or rural-specific purposes (e.g., Sianipar et al.'s Design Methodology for Appropriate Technology; Pearce's Open-Source Appropriate Technology).
  • By following a formal engineering design method, the authors can argue how they methodically identify and solve precise problems through part-by-part of the system's structure. The performance of the proposed water supply system might be testable empirically; however, without a formal engineering design, the proposed design of the water supply system cannot be proven scientifically.

Response: We thank you for your useful comments. We have revised the literature review in the introduction section (lines 54-127) with about 25 new references. According to your suggestion, we also added a new paragraph regarding to engineering design (lines 192-201).

Round 2

Reviewer 4 Report

After a thorough check on the revised manuscript, I see that the authors put effort into revising their manuscript. However, the revisions/additions have not been adequate to address concerns raised in the previous review round. I suggest the following revisions for this second review round.

LITERATURE REVIEW.

  • The revisions applied in the "Introduction" section have not addressed the concern of a literature review to develop the underlying structure of the technical design of the proposed water supply system. Without a rigorous literature review on the necessary structure and components/sub-assemblies, the structural design of the proposed water supply system is arbitrary and scientifically baseless.
  • I suggest the authors add the conceptual structure of the technical design by following a common conceptual design approach (for example, design black-box ⇒ DOIs 10.1080/0954482090309957; 10.1017/S0890060409990163). The literature review is necessary to support arguments when establishing the technical structure and components/sub-assemblies. In practice, the literature review should gather the structural components necessary to build the intended supply system of drinking water based on harvested rainwater.
  • Please put the literature review section in between the "Introduction" and "Materials and Methods" sections.

ENGINEERING DESIGN.

  • I see that the authors have decided (lines 166-175) to use an established engineering design for this research (Sianipar et al.'s). However, the technology development in this research has not followed the design processes/stages guided by the chosen methodology. I am afraid that the choice is a mere decision to put an engineering design methodology into the manuscript.
  • In general, I recommend the authors check seminal books in engineering design (for example, Samuel & Weir's "Introduction to Engineering Design") and some examples of using engineering design methodologies in technology development (for example, DOI: 10.3390/s22062108, pages 5-7).
  • As for the selected engineering design in this research (Sianipar et al.'s), I suggest the authors rearrange their design processes/stages to follow the design flow suggested by the methodology. If the authors intend to apply the methodology partially, I recommend the authors check examples of its partial uses (for example, 10.1016/j.techsoc.2021.101658).

Author Response

This manuscript is a resubmission of an earlier submission. The following is a list of the peer review reports and author responses from that submission.

Round 1

Reviewer 1 Report

The paper has improved after taking into account the reviewers comments.

Nevertheless, the provided assumptions for drinking water demand in rural areas of developing countries (line 287 to 303) do not include any information how water needed for hygiene & cleaning is provided and makes the reader think safe drinking water is only demanded for drinking and cooking. This does not follow the UN requirements regarding the right to water - https://sr-watersanitation.ohchr.org/pdfs/faq.pdf

The 20l/day can work for drinking, cooking, hygiene and washing dishes & laundry in poor rural areas if water is reused within the household, e.g. reuse of water after dish washing for floor cleaning,.... 20l/d are given by the UNHCR as necessary assumption when designing long term water supply in refugee camps. 10l are only recommended for short term water supply.

The UN recommends 50 l/day ; The WHO 25l/day - taking into account 10/day as a scenario does not make sense when designing and calculating BCR for RWH systems which are definitely designed and LCC ist calculated for long term water supply.

therefore the paper has to include an example for a 25l/d and a 50l/d scenario.  provide sound conclusions from the results. it can be discussed that 50l/d are not be feasible if reliability and LCC are taken into account. It would be of high value for the scientific community to read about thresholds of water demand, community sizes, hydrology, ...... if RWH systems for drinking water supply are installed while costs have to stay low 

further, it is not understandable why larger tank sizes result in a region where the rainfall is higher (Vietnam) as described in line 476 to 479 ?? it is state of the art to install an overflow for times when rainfall is higher than consumption. the larger tank sizes in my point of view can only result from the lower number of roofs (houses) for the same number of inhabitants respectively consumption. please clarify this issue.

Reviewer 2 Report

Re-submitted paper, the main problem is still in, and has not been well modified. 

Reviewer 3 Report

The paper addresses a topic of interest to the scientific community related to alternative sources of water supply. The paper could be improved to achieve a greater impact:

1- Title: It is suggested to include that it is a comparison of two case studies. Currently, it is generalized from two specific cases.

2- The abstract should be improved by including some methodological details (how was the water balance done?). Likewise, it must conclude by showing the implications of the results obtained.

3- In the introduction (lines 79 - 85) it could be mentioned that several RWH projects have been carried out at the community level but these are not sufficiently documented and many times they have been carried out empirically.

4- In the introduction, show more clearly the scope of the work.

5 - The materials and methods section should be rewritten to improve its structure (eg item 2.3. Methods has the same name as Item 2. Materials and methods). In some sections it is excessively extensive, which can lead the reader to lose the focus of the work. It is suggested to rely on supplementary material and synthesize some aspects (eg description of the technology and the Reliability analysis).

6 - Some questions in the methodology are:
a) how to determine that the two locations (Australia and Vietnam) are representative?
b) How was this technological option for water treatment selected? Real information on water quality is not necessary in each context? or at least a literature review with some data?
c) Are operating costs of the system included? I have seen maintenance, investment and replacement.
d) Lines 423 and 448 seem not to be methodological. Could you be part of the results discussion?

7 - Some elements raised in the conclusions could be part of the discussion. Instead, in the conclusion highlight the most relevant points of the study.